# Is there meaningful influence from situational and environmental factors on the physical and technical activity of elite football players? Evidence from the data of 5 consecutive seasons of the German Bundesliga

**Paweł Chmura**[1], **Hongyou Liu**[2], **Marcin Andrzejewski**[3], **Jan Chmura**[4], **Edward Kowalczuk**[5], **Andrzej Rokita**[1], **Marek Konefał**[4]*

**1** Department of Team Games, University School of Physical Education, Wrocław, Poland, **2** School of Physical Education & Sports Science, Guangzhou Higher Education Mega Centre, South China Normal University, Guangzhou, China, **3** Department of Methodology of Recreation, Poznań University of Physical Education, Poznań, Poland, **4** Department of Biological and Motor Sport Bases, University School of Physical Education, Wrocław, Poland, **5** Football Club, Hannover 96, Hannover, Germany

* marek.konefal@awf.wroc.pl

**Data Availability Statement:** The data that was used for this study was acquired from a thirdparty,

## Abstract

The study aimed to identify the effects of situational (match location, match outcome and strength of team/opponent team) and environmental (ambient temperature, relative humidity, WBGT, ground and weather condition) factors on the physical and technical activity of elite football on individual playing positions. Physical and technical activity were collected from 779 football players competing in the German Bundesliga during 5 domestic seasons, from 2014/2015 to 2018/2019, totalling 1530 matches. The data on players' physical and technical activity was taken from the IMPIRE AG system. Based on the available data, 11 variables were selected to quantify the match activity profiles of players. The results showed that situational variables had major effects on the technical performance (especially number of passes performed) but minor effects on physical performance. In turn, among the analysed environmental factors, temperature is the most sensitive, which affects the Total Distance and Sprint Efforts of players in all five positions. This investigation demonstrated that, given that passing is a key technical activity in modern football, players and training staff should be particularly aware that passing maybe affected by situational variables. Professional players are able to react and adapt to various environmental conditions, modifying physical activity depending on the needs in German Bundesliga. These results could help coaches and analysts to better understand the influences of situational and environmental variables on individual playing positions during the evaluation of players' physical and technical performance.

formerly http://www.offizielle-spieldaten.de/, now https://matchanalysishub.bundesliga.com/login. The data was provided under a scientific cooperation with a football club currently appearing in the 2nd Bundesliga. The authors' ethical approval also prevents them from sharing any data in any way that could be re-identified. The metadata and the (score) data itself would allow someone else to re-identify teams and possibly players. However, access to the data should be possible from the third-party. The data acquired were so called 'excel dumps' of player statistics per match of the German league from season 2014/15 to 2018/2019. Access to the data can be organised by contacting Match Analysis Hub: mdc@sportec-solutions.de.

**Funding:** The authors received no specific funding for this work.

**Competing interests:** Authors declare that they do not have any commercial affiliation relating to employment, consultancy, patents, products in development, or marketed products, etc. This does not alter our adherence to PLOS ONE policies on sharing data and materials.

# Introduction

The high dynamic and multidirectional character of modern-day football matches requires detailed match analysis with state-of-the-art, technologically-advanced motion analysis systems [1]. Furthermore, the high level of validity and reliability of these systems allows for many possible applications in scientific research connected to the physical [2] and technical activity [3], as well as the tactical play [4] of professional football players. In turn, Paul et al. [5] recommended introducing changes in the design of time–motion studies, notably using a more holistic approach to monitoring and larger sample sizes. Any holistic approach must also take into consideration the performance-related and situational factors recognised to affect physical and technical activities of football players [6]. Moreover, Memmert et al. [7] stated the investigation of physical and technical parameters provides an objective understanding of actual match performance and can help to explain the differences between successful and unsuccessful match performances.

Many authors emphasise that football players' in-game physical and technical activities are influenced by different situational factors and/or contextual variables [8, 9]. It has been previously reported that match location (home advantage), match outcome (match result) and strength of team and opposition are deemed the most significant situational factors influencing team performance during a football match [10, 11]. Firstly, there is a tendency for teams that play at home to score more goals [12], perform more shots on goal, more crosses, more passes, more successful passes, and more successful dribbles compared with teams playing away [10, 12–14]. Secondly, analyses based on match outcome, with a consideration of the players' field position, can be very useful in explaining differences in the impact of situational factors on physical and technical activities of players [11, 15]. In the German Bundesliga, in won matches, it has been observed that, players in offensive positions ran a significantly longer distance especially for distances covered at intensities of 21–23.99 and above 24 km/h [16, 17] Thirdly, evidence indicates that stronger teams: a) dominated ball possession against their opponents [11, 18, 19], b) demonstrated more stable patterns of play, independently of the evolving score-line [11, 13], and c) did not experience the same home advantage as inferior opponents [14].

As football is an ever-changing game, performed in a range of environmental conditions, understanding the effects of factors such as ambient temperature, and relative humidity is required to optimise player performance [20]. Environmental conditions, beyond match location, strength of team, and match outcome, is one of the most important influencing variables that affect the physical activity of players in football matches of high competition level [21]. In previous studies, Maughan et al. [22] observed that players attained the highest level of physical activity in the 4–10˚C temperature range. According to Grantham et al. [23], playing football at an air temperature under 22˚C is not a risk for heat stress and its negative consequences, while it is at low risk between 22 and 28˚C, but high risk above 28˚C. Link and Weber [24] reported players in the Bundesliga reduced their total distance when playing in high temperatures, whilst maintaining their ability to perform high-intensity actions when required. This fact was confirmed by studies of Chmura et al. [25] who, examining the physical activity profile of players at the World Cup in Brazil, revealed that in hot environments, players preserved key physical performance measures (e.g., peak sprint speed) that are associated with match outcome [26] by reducing the number of sprints, but simultaneously improved the rate of successful passes performed during a match [27]. An additional burden for football players' bodies exposed to high ambient temperatures is high relative humidity. In such conditions the elimination of body heat by evaporation is seriously hindered [23, 28]. Research shows that at lower relative humidity, humans show improved cardiorespiratory, thermoregulatory, and perceptual responses during exercise than they do at higher humidity levels [29]. In turn, after exceeding the 61% relative humidity threshold, the time to exhaustion of the incremental exercise test and match performance were significantly reduced [25, 30].

Thornes [31] already stated that "not only are temperature and humidity important in comprehending the impact of weather on match performance in different sport, but also wind velocity, visibility, precipitation, sunshine and the state of the ground". Hence, this too, has also been taken into consideration, as well as more complex indicator of the average impact of heat on humans—Wet Bulb Globe Temperature (WBGT) index, which also accounts for the level of sunlight and wind strength [32]. Analysing the changes in WBGT, it has been observed, that in team sport, players can modulate their activity in high or extreme environmental conditions during the matches, with similar trends as those for temperature and humidity [33, 34]. Recently, similar research has been published by Zhou et al. [35], who analysed the physical and technical activity of teams, albeit only in one season of the Chinese super league of football, with a smaller number of climatic conditions and did not take into account the diversity of players in positions.

To the best of the authors' knowledge, ground conditions (sometimes also known as quality of surface or level of surface, wear: no-wear/low-wear/average-wear/heavy-wear) and weather conditions (sunny/light-rain/heavy-rain/snow) affect the physical and technical activity of football players has not been fully analysed so far. Bad ground and weather conditions may lead to fatigue; for example, running on a muddy ground is more exhausting [36]. Moreover, adverse conditions such as wind and rain may affect playing skills because of dirty balls, muddy pitches, and wind-affected kicks [37], thus affecting the style of play and the quality of the game directly, through significant decrease in the number of successful passes, and significant increase of unsuccessful passes and interceptions [38]. It appears that it is pertinent for practitioners and policy makers alike to investigate the effect of these conditions. Further understanding of the effect of these differentiated conditions on physical and technical activity of players would allow better planning to minimise possible detrimental factors [20, 39]. It is also particularly important, because of the fact that, as the climate becomes warmer, more and more matches are being played in extreme environmental conditions and players are increasingly more often exposed to adverse weather conditions.

Therefore, the current study aimed to identify the effects of situational (match location, match outcome, and strength of team/opponent team) and environmental (ambient temperature, relative humidity, WBGT, ground and weather conditions) factors on the physical and technical activity of elite football. This comprehensive analysis, based on a large dataset in one of the best European leagues, aims to provide important insights into the highly dynamic and complex nature of a football match on individual playing positions.

## Materials and methods

### Sample and variable

Physical and technical activity data were collected from 779 football players competing in the German Bundesliga during 5 domestic seasons, from 2014/2015 to 2018/2019, totalling 1530 matches. This study maintains the anonymity of the players following data protection law, is conducted in compliance with the Declaration of Helsinki and was approved by the Senate Committee on Research Ethics at the University School of Physical Education in Wrocław (no. 20/2017).

Based on the availability of data and previous literature [14, 18, 21, 40–42], 11 physical and technical activity-related match events and actions were selected as variables to identify the physical and technical match performance of players. Detailed information of these variables can be found in Table 1. Complete definitions of physical and technical variables are available at the Deutsche Fußball Liga (DFL) [43]. Definitionskatalog Offizielle Spieldaten–Bundesliga website. https://s.bundesliga.com/assets/doc/10000/2189_original.pdf.

**Table 1. Categories of situational and environmental (independent) variables and performance indicators (dependent variables) and its definition and/or collection procedures.**

| | Variables | Definition and/or collection procedures |
|---|---|---|
| Independent | **Situational** | |
| | • Match location | Recorded as "home" or "away" depending on whether the sampled team was playing at its own ground or that of its opponent. |
| | • Match outcome | Matches were defined as "won", "drawn" or "lost" in relation to the number of goals scored and conceded by a team. |
| | • Strength of team | The end-of-season rank of the sampled team. |
| | • Strength of opposition | The end-of-season rank of the opponent team. |
| | **Environmental factors** | |
| | • Temperature | Temperature is the indication of the temperature in °C approximately 30 minutes before the start of the game in the stadium and can be obtained from the data collector via a mobile weather station at the recording site. |
| | • Humidity | Air humidity is the indication of the air humidity in % approximately 30 minutes before the start of the game in the stadium and can be obtained from the data collector via a mobile weather station at the recording site. |
| | • WBGT | The wet-bulb globe temperature (WBGT) is a type of apparent temperature used to estimate the effect of temperature, humidity, wind speed (wind chill), and visible and infrared radiation (usually sunlight) on humans. |
| | | WBGT was calculated as WBGT = 0.7 wbt+0.2 bgt+0.1 dt; where wbt is wet bulb temperature, bgt is radiant heat and dt is dry bulb temperature. |
| | **Ground condition** | |
| | | Ground conditions is the degree of wear of the surface at the beginning of the game, as determined by the game data collector. This either shows no wear (no space wear), smaller worn surfaces (light space wear), larger worn surfaces (medium space wear) or large areas of worn surfaces (heavy space wear). |
| | **Weather condition** | |
| | | Precipitation is an indication of whether and if so how much it rains or snows on the field about 30 minutes before the start of the game. If this information changes significantly after the time of recording, the expression of the precipitation present for the predominant playing time was specified, or an expression which, according to the game data creator, had the greatest influence on the game. |
| Dependent | **Physical activity** | |
| | • Total Distance (TD) | Total distance covered by a player during match play |
| | • Top Speed | Maximum running velocity of a player during match play |
| | • Sprint Effort (SprintE) | Efforts of sprint (velocity > 5 m/s) achieved by a player during match play |
| | • High Intensity Effort (HIE) | Efforts of run (velocity > 4 m/s) achieved by a player during match play |
| | **Technical activity** | |
| | Shot | An attempt to score a goal, made with any (legal) part of the body, either on or off target. |
| | • Pass | An intentional played ball from one player to another. |
| | • Duel | The action of gaining possession from an opposition player who is in possession of the ball. |
| | • DuelWon | A won duel means the player beats the defender while retaining possession |
| | • DuelLost | A lost duel is one where the dribbler is tackled and loses possession. |
| | • DuelSucc | Successful duels as a proportion of total duels. |
| | • Cross | Any ball sent into the opposition team's area from a wide position. |

## Data resource

The data on players' physical and technical activity was taken from the IMPIRE AG system (Ismaning, Germany). The system generates official DFL reports, which were then analysed and interpreted. The analysis was carried out using an IMPIRE AG motion analysis system [44], providing records of all players' movements, with a sampling frequency of 25 Hz. IMPIRE AG and Cairos Technologies AG (Karlsbad, Germany) provides a ready to use vision-based tracking system for team sports called VIS.TRACK. That system consists of two cameras and offers software tracking of both players and the ball [24]. The major advantages of vision-based systems lie in their high update rate corresponding to the camera frame rate, and the

fact that the players and the ball are tracked simultaneously, i.e., each position sample for a single player has a corresponding position sample for every other player and the ball, measured at the identical point in time [45]. The validity and reliability of this system for taking such measurements have been described in detail elsewhere [24, 44, 46, 47].

## Statistical analysis

A generalised mixed linear model was realised with Proc Glimmix in the University Edition of Statistical Analysis System (version SAS Studio 3.6). A random effect for player identity was used to account for repeated measurement on the players. The fixed effects estimated the effect of situational and environmental factors. Separate Poisson regressions were run in the model, taking the value of each of the 11 physical and technical activity-related parameters as the dependent variable. The results of each modelling were output by players' positional group.

Game location and match result were included as nominal variables with two levels (home & away, and won & not-won, respectively). The effect of team strength and opponent strength was estimated by including the difference in the log of the end-of-season ranks as a predictor [35, 48]. The ground and weather conditions were added as four-level nominal variables (no-wear/low-wear/average-wear/heavy-wear, and sunny/light-rain/heavy-rain/snow, respectively). Temperature, humidity and WBGT were included as numeric linear effects using their raw values, and their magnitudes were quantified as the effect of two of their standard deviation: [49] the predicted value for a typically high value of the predictor (1SD above the mean) minus that for a typically low value (1SD below the mean). The raw value of temperature and WBGT was included as a quadratic effect to allow for the possibility and estimation of an optimum temperature/WBGT defined by the maximum value of the quadratic. If the maximum occurred within the range of environmental temperatures/WBGT, its confidence limits were derived by parametric bootstrapping [50]. The optimum temperature/WBGT and confidence limits were reported if at least 90% of the 10,000 bootstrapped samples produced a maximum; otherwise, it was evident that the effect of temperature/WBGT was approximately linear and was therefore estimated and reported as the effect of two standard deviations.

Observed linear effects and their 99% compatibility intervals were assessed and presented in standardised units, whereby the change score or difference in means was divided by the observed between-player standard deviation (SD) derived from the mixed model, and then evaluated qualitatively with the following scale: <0.2 trivial, 0.2–0.6 small, 0.6–1.2 moderate, 1.2–2.0 large, >2.0 very large. Decisions about magnitudes accounting for the uncertainty were based on hypothesis tests for substantial and trivial effects [51]. Hypotheses of substantial decrease and increase, or negative and positive difference were rejected if their respective p values (p– and p+) were less than 0.05. If only one hypothesis was rejected, the p value for the other hypothesis corresponds to the posterior probability of the magnitude of the true effect in a reference-Bayesian analysis with a minimally informative prior [52], so it was interpreted with the following scale: >0.25, possibly; >0.75, likely; >0.95, very likely; >0.995, most likely [49]. If neither hypothesis was rejected, the effect is described as unclear. For the purpose of being more vigilant, only the very likely and most likely effects are to be discussed in this study.

## Results

### Descriptive statistics

A total of 21,971 individual match observations (Central Defender—CD n = 6187, Full-Back—FB n = 4806, Central Midfielder—CM n = 4798, Wide Midfielder—WM n = 3545, Forward—FW n = 2635) were made for outfield players (goalkeepers excluded). Mean body height among players was 183.21 ± 6.59 cm, mean body mass 78.33 ± 7.28 kg, and mean age

**Table 2. Descriptive statistics.**

| Independent variables | | | | | |
|---|---|---|---|---|---|
| | | Mean | SD | Min. | Max. |
| Temperature | | 11.9 | 7.1 | -7.0 | 35.0 |
| Humidity | | 62.4 | 19.2 | 0.0 | 100.0 |
| WBGT | | 14.5 | 5.2 | 1.4 | 30.9 |
| Dependent variables (mean ± SD) | | | | | |
| | CD | FB | CM | WM | FW |
| Physical | | | | | |
| TD | 10.1±0.6 | 10.8±0.6 | 11.6±0.7 | 11.3±0.8 | 10.8±0.9 |
| Topspeed | 30.6±1.7 | 31.5±1.4 | 30.2±1.6 | 31.6±1.5 | 31.4±1.5 |
| SprintE | 13.3±5.2 | 22.9±6.4 | 17.1±6.6 | 25.8±7.5 | 24.4±7.4 |
| HIE | 31.3±8.3 | 41.3±9.2 | 48.2±12.0 | 47.7±10.7 | 44.1±11.2 |
| Technical | | | | | |
| Shot | 0.5±0.8 | 0.6±0.9 | 1.1±1.3 | 1.8±1.6 | 2.6±1.8 |
| Pass | 51.0±24.0 | 45.0±18.0 | 48.0±21.0 | 35.0±15.0 | 26.0±11.0 |
| Duel | 15.1±5.6 | 17.6±6.1 | 21.6±7.0 | 22.4±7.5 | 24.6±8.5 |
| DuelWon | 9.1±4.0 | 9.6±4.1 | 11.0±4.3 | 10.5±4.3 | 10.6±4.6 |
| DuelLost | 6.0±3.0 | 8.1±3.5 | 10.6±4.4 | 11.8±4.7 | 14.0±5.4 |
| DuelSucc | 60.0±15.0 | 54.0±13.0 | 51.0±12.0 | 47.0±12.0 | 43.0±11.0 |
| Cross | 0.1±0.5 | 2.4±2.4 | 0.8±1.5 | 2.2±2.4 | 0.9±1.5 |

26.29 ± 3.41 years. Descriptive statistics of physical and technical match performance-related parameters for players of five positions are presented in Table 2.

## Effects of situational factors

Fig 1 presents the effects of situational variables on the match performance-related parameters. For the effects of match location, CD and FB achieved substantially higher values in passing when playing at home than when playing away. For the effects of match result, CD, FB and CM made more passes in games won than in games not won. For the effects of team and opponent strength, in the situation of a stronger team playing against a weaker opponent, CD achieved more passes, duels and duels won, FB made more passes and crosses, CM achieved higher values in pass but lower values in TD and HIE, WM made more passes, crosses and shots, while FW achieved higher numbers of shots but lower numbers of duels and duels lost.

## Effects of environmental factors

No clear optimum temperature/WBGT was detected in the bootstrapping, hence the effect of temperature/WBGT was approximately linear. As can be seen from Fig 2, TD and HIE of players from all the five positions decreased when the temperature increased, meanwhile, SprintE of FW and Duel of CM showed the same trend. All the physical and technical performance-related parameters showed trivial changes when the humidity and WBGT increased by two standard deviations. Fig 3 presents the influence of ground condition on the match performance of players. As be shown, there were no meaningful differences detected in the performance in matches played on low-wear and average-wear ground vs on no-wear ground, while duels won by WM in matches played on heavy-wear ground was higher than on no-wear ground. Effects of weather condition on the match performance of players are displayed in Fig 4. As can be seen, there was no meaningful difference found in the performance in matches

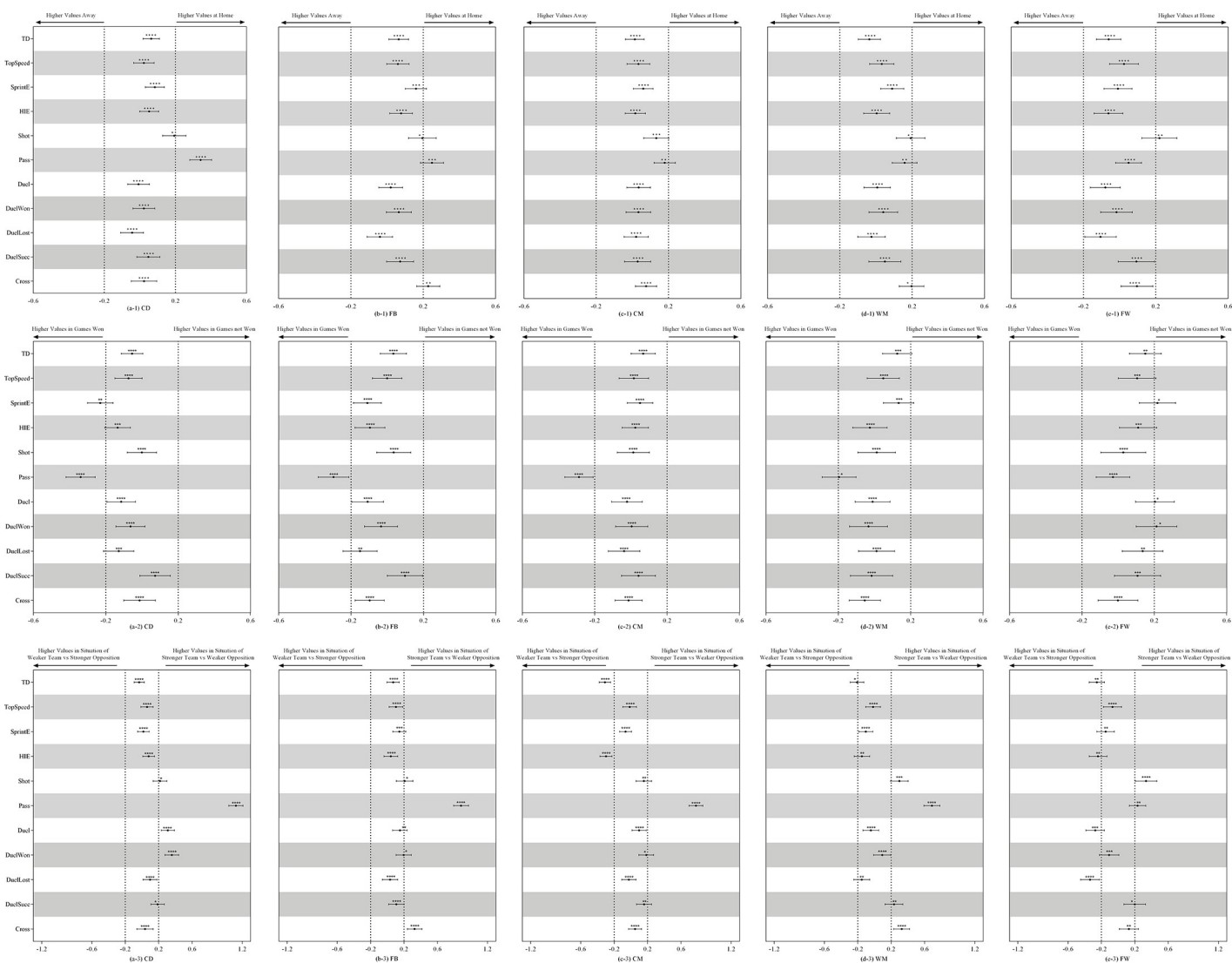

**Fig 1. Effects of situational variables on the technical and physical performance of players of different positions.** Effects of match location and match result are shown as the standardised difference in each performance-related parameter when playing at home vs playing away, and in matches won vs not-won. Effects of team and opponent strength are shown as the effect of an increase of two standard deviations in the log of rank difference on the standardised change score of each performance-related parameter. Bars are 99% compatibility intervals. Dotted lines represent the smallest worthwhile difference/change. Asterisks indicate the likelihood for the magnitude of the true effect as follows: * possible; ** likely; *** very likely; **** most likely. Asterisks located in the trivial area denote the likelihood of trivial effects.

played in light rain vs sunny conditions, while, in the snowing conditions, CM made fewer duels and WM achieved high duel success.

## Discussion

The study aimed to identify the effects of situational and environmental factors on the match performance of elite football considering positional differences. Our main findings include that: (1) situational variables had major effects on the technical performance (especially number of passes performed) but trivial effects on the physical performance; (2) among the analysed environmental factors, temperature is the most sensitive one, which affects the TD and SprintE of players of all the five positions.

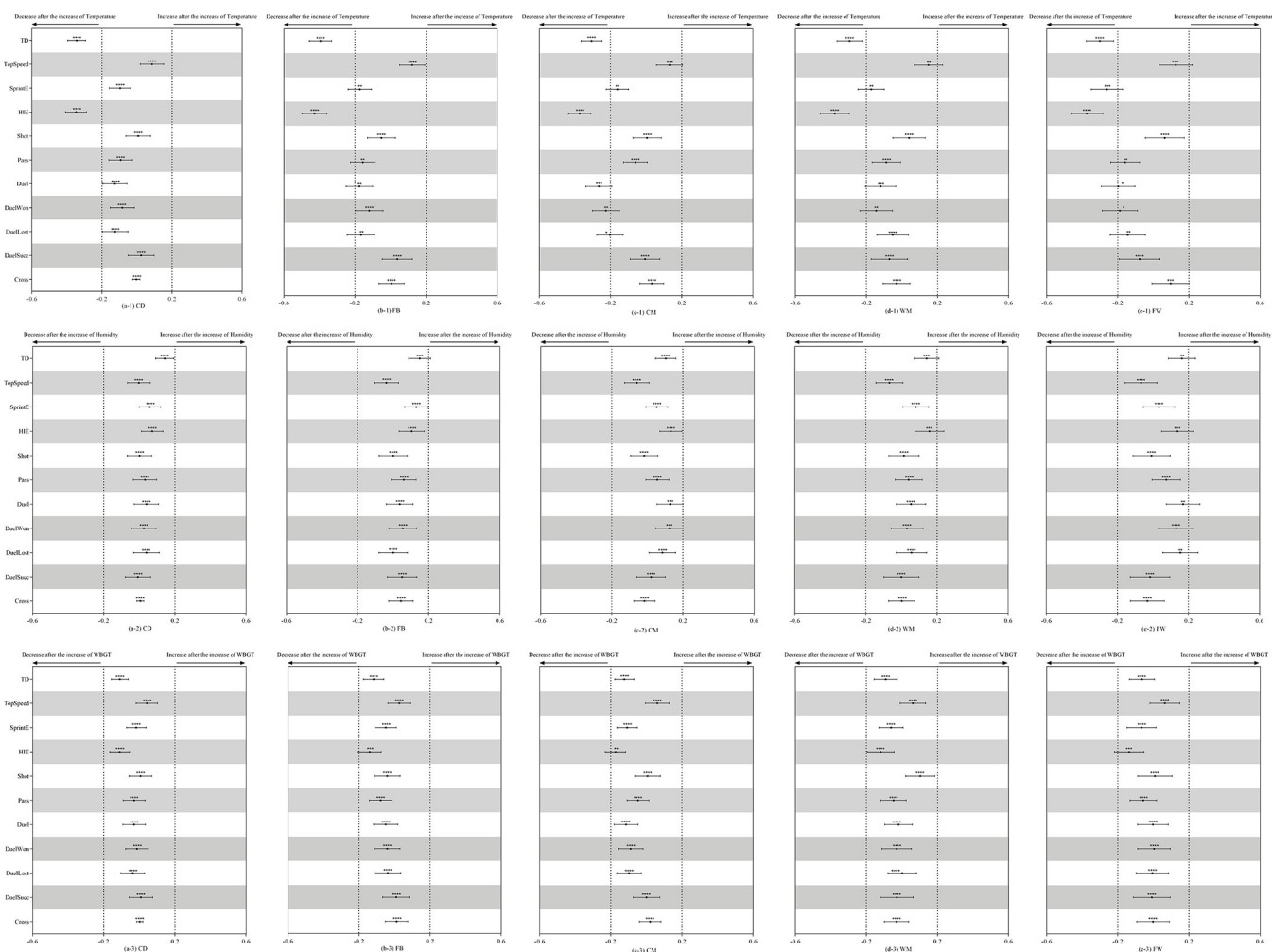

**Fig 2. Effects of WBGT and humidity on the technical and physical performance of players of different positions.** Effects are shown as the effect of an increase of two standard deviations in the value of WBGT and humidity on the standardised change score of each performance-related parameter. Dotted lines represent the smallest worthwhile change. Asterisks indicate the likelihood for the magnitude of the true effect as follows: ** likely; *** very likely; **** most likely. Asterisks located in the trivial area denote the likelihood of trivial effects.

Home advantage in football has been discussed in depth and it is believed to affect the choice of tactic and strategy in competition, hence has an influence on the technical and physical match performance of players [9, 14, 53]. Based on the data of team performance in the Chinese Super League of football, Zhou and colleagues [35] pointed out that the influence of home advantage mainly occurs in technical activities rather than physical ones. Accordingly, our results showed that all the analysed four physical performance-related parameters achieved by players of all five positions in home matches were similar to the away matches in the Bundesliga. Furthermore, all the duel-related actions (duels made/won/lost and duel success) performed by players of different positions in home and away matches were analogous. The most notable difference occurred in the number of passes made by CD and FB. As one of the most frequent technical activities in football, around 500 passing interactions are normally achieved by a single team during a game [54]. Król et al. [55] observed that currently there is a tendency where many teams decide to adopt a strategy of short passes, which makes it possible to minimise the risk of losing the ball and, at the same time, increases ball possession during the game

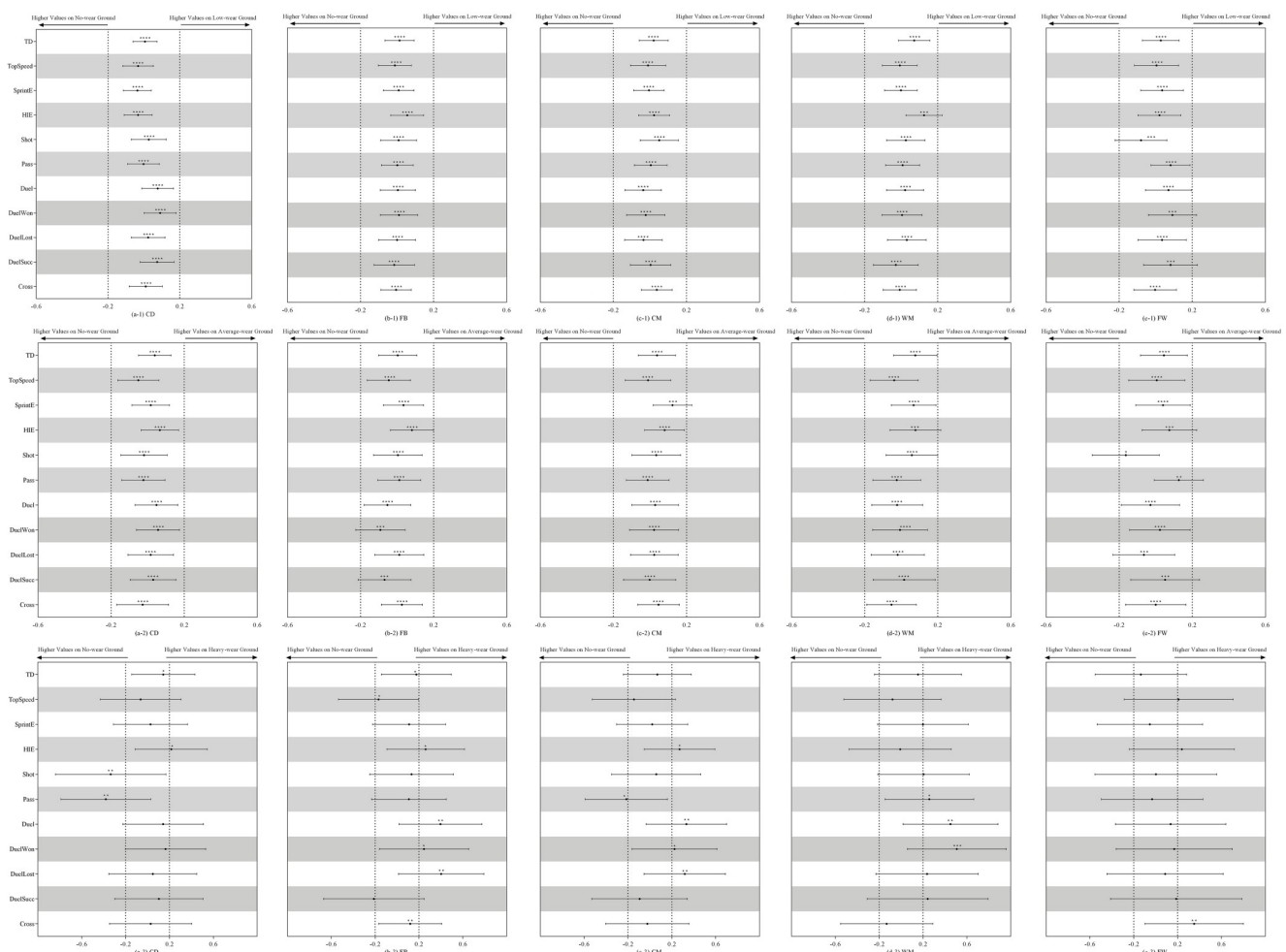

**Fig 3. Effects of ground condition on the technical and physical performance of players of different positions.** Effects are shown as the standardised difference in each performance-related parameter when playing on low-wear, average-wear, and heavy-wear ground vs playing on no-wear ground. Bars are 99% compatibility intervals. Dotted lines represent the smallest worthwhile difference. Asterisks indicate the likelihood for the magnitude of the true effect as follows: * possible; ** likely; *** very likely; **** most likely. Asterisks located in the trivial area denote the likelihood of trivial effects.

[56]. Barnes et al. [40] indicate that defenders are becoming more and more involved in the passing and possession of the ball in modern football matches. Our findings tend to show that defenders in their home stadiums are even more likely to possess the ball and lead a positional attack than playing on away pitches. It is also worth noting that home and away differences in Shots for CD, FB, WM and FW, and Cross for FB and WM are possibly or likely to be substantial, which calls for further investigation.

Taking into account the second analysed situational factor—match outcome, it has been found that players in matches won perform more physical activity in terms of the sprints and high-intensity efforts in the German Bundesliga [2, 16, 57]. In won matches, players are also characterised by achieving a greater number of technical activities, such as performing more shots, more shots on target, more shots from both the penalty area and from outside the penalty area, and making more passes [16]. In comparison to prior research, our analysis showed

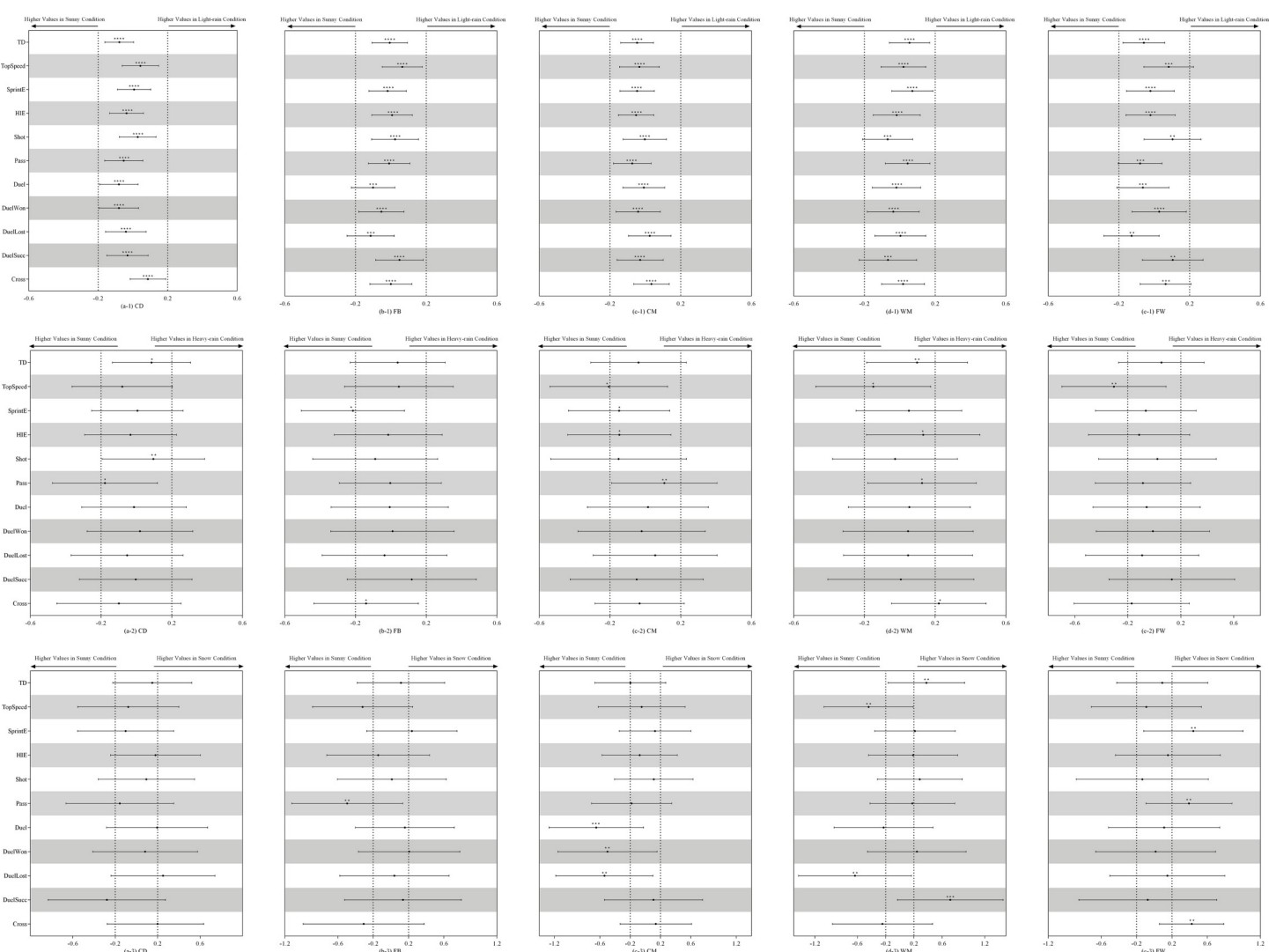

**Fig 4. Effects of weather condition on the technical and physical performance of players of different positions.** Effects are shown as the standardised difference in each performance-related parameter when playing in light-rain, heavy-rain, and snowy conditions vs playing in sunny conditions. Bars are 99% compatibility intervals. Dotted lines represent the smallest worthwhile difference. Asterisks indicate the likelihood for the magnitude of the true effect as follows: * possible; ** likely; *** very likely; **** most likely. Asterisks located in the trivial area denote the likelihood of trivial effects.

that in won matches, players did not confirm high physical activity in any position. This is quite surprising because the requirements with regard to fast play in modern football for professional players are becoming more and more demanding [2, 17, 58]. On the other hand, we have observed that defenders and CM performed more passes. In football matches, approximately 80% of goals scored result from 3 or more consecutive passes [59, 60]. The technical activity of passing is a key skill underlying successful performance in football [61, 62] and depends on many conditions, including a player's scoring status, the opponent's defensive pressure or a team's specific style of play [63]. Fast transition of the ball to offensive areas of the pitch through a combination of high pass rates and ball speed is advantageous in elite football [64], and have a strong association with success [62]. Furthermore, Barnes et al. [40] found that, players in the English Premier League performed ~40% more passes, with a greater percentage of successful passes occurring in 2012–13 (84%) compared to 2006–07 (76%). In

turn, Konefał et al. [16] indicate that the number of passes made by CD increased in matches won, and by CM and WM players in all matches, whereas pass accuracy increased at the CM position in won and drawn matches. Our research emphasises that if a team would like to win the match, CD, FB and CM position players would need to achieve an increased number of passes.

Effects of the strength of team and opposition on the variation of technical performance of players is greater than the effects of match location and match outcome, as reported by Liu et al. [18], analysing data of the Spanish First Division Professional Football League (La Liga). It has been found that, the stronger teams covered more total distance and high-speed-running distance whilst in possession of the ball [20] and made more attacking related actions (shots, shots on target, crosses, passes, passing accuracy), as well as less defensive actions (tackles, yellow cards) than weaker teams [14, 35, 65]. In line with these studies, we identified a decrease in the total distance and distance with high intensity by CM in the situation of a stronger team playing against a weaker opponent. According to Nassis et al. [33] this adjustment of players' behaviours raises the percentage of successful passes and key passes. However, our findings also reveal that, CD, FB, CM and WM make more passes when playing in a game with higher disparity (stronger team vs weaker opposition). Passing parameters are key variables to distinguish different tactical approaches [66] and in creating goal scoring opportunities [67]. Indeed, some teams try to create opportunities by long, direct passes, whereas other teams have a more elaborate possession-style of play. Analysing individual positions, this situational factor leads to more duels performed and won by CD. This result is contrary to what has already been established in the literature, where players try to decrease the unnecessary risk of losing the ball, especially in these positions [53]. In turn, players in lateral positions–FB and WM–performed more crosses. This finding demonstrates that crossing is a main technical requirement for these positions and one of the important ways to create situations to score goals [59]. In WM and FW positions, the most important activity is to take a large number of shots, which concurs with the other authors [18, 65]. In addition, when a team is stronger than their opponent, duels and lost duels performed by FW decline. This result also contradicts those obtained by Konefał et al. [53] in the Bundesliga, where players in this position in close games performed more technical and physical activity in won matches. The difference may be due to the fact that in these studies, all matches with a greater goal advantage were considered, so the forwards could be less motivated to be more active in the match.

In this work, apart from situational factors, the impact of environmental factors was examined. In the existing literature, it is considered that professional players in unfavourable environmental conditions reduce their high-intensity activity in order to remain effective in technical activities [27, 33]. Moreover, many authors indicate the "comfort zone" to promote football performance and higher or lower temperatures may impair physical activity in football [25]. In our research it was reported that increased temperature will lead to the reduction of total distance and high-intensity effort in all positions and decrease the number of sprinting efforts by FW. The number of duels also decreases for CM. Similar observations have been registered by Zhou el al. [35], who observed the major effects of environmental factors on physical activity and only a minor influence on technical activity in the Chinese Super League. This result also shows that players nowadays not only limit high-intensity activity, but also overall activity related to total distance covered, when playing matches in higher ambient temperature. Despite this, in the present study, we did not observe the expected influence of humidity and WBGT on changes in physical and technical activity in professional football players performing in Bundesliga. Players and coaching stuff used different heat and cold alleviation strategies, playing in unfavourable conditions and they know how to adapt to the more unfavourable conditions [34]. This may also be due to a large proportion of these matches

being played in moderate conditions and that the studied players in this league represent the highest European level.

With regard to ground and weather conditions, so far, this topic has not received much attention in football. Recently, only studies on the number of accelerations [68, 69] and the reaction of the ground to player performance have been published [70–72], albeit more from a biomechanical point of view. However, its influences on real-game physical and technical activity are often overlooked. Although nowadays the performances of professional players on worse surfaces occur much less frequently, ground conditions are worth investigating. It was assumed that lowering the quality of the ground (low and medium wear) on which the players operate in comparison to no wear, would negatively affect physical and technical activity, but none of these assumptions were proven correct. Only minor changes have been detected. Another area of research was to ascertain how weather conditions affect players' physical and technical activity. In this case, not many large-sample studies were found on identifying which weather types are the best or worst for performing physical and technical activity in football matches. Similar with the effects of ground conditions, not many substantial effects of deteriorating weather conditions were detected, apart from midfield positions; in the case of playing in snow CM perform fewer duels while WM perform more. It seems that such conditions additionally intensify the specificity of tasks in these positions. These results show that the players in the Bundesliga do not modify their activity in relation to various ground and weather conditions.

It is worthwhile noting that data from only one league was taken into account in the current study. In further research, the situational and especially environmental factors in other European leagues consisting of larger research samples, (e.g. English Premier League, French League 1, Italian Serie A, Spanish La Liga) should be examined. In subsequent analyses, in addition to taking into account the situational factors, attention should also be paid to close games as well as only matches played in unfavourable environmental conditions. Future studies are also recommended to analyse more parameters of physical and technical activity.

## Conclusion

Our study demonstrated that situational variables had major effects on the technical activity, but trivial effects on the physical activity in German Bundesliga players. Players performing at their home stadium, on winning teams and on teams stronger than the opponent, make more passes at individual positions. This indicates that professional football players and training staff should be particularly aware that passing is a key technical activity in modern football, which is sensitive to different kinds of situational variables.

From the range of environmental factors tested, only temperature affects physical activity, especially on total distance and number of sprints, whilst only trivial effects were observed on technical performance in this league. Decreasing and/or increasing the humidity and WBGT beyond the comfortable range does not affect physical and technical activity. Similarly, the deterioration of ground wear and the deterioration of weather conditions do not show substantially negative effects, apart from snowfall which affects midfield positions in terms of the number of duels. This indicates that professional players in the German Bundesliga do not modify and/or adopt their behaviour quickly with respect to different environmental conditions.

The information from our study may help coaches and analysts to better understand the influences of situational and environmental variables on individual playing positions during the evaluation of players' physical and technical performance. All the above-mentioned factors are particular challenges for football coaching staff in their efforts to ensure optimal physical

and technical preparation of the players. The research underlines that training staff should develop the appropriate competences needed to carry out integrated data analysis, whilst also taking into account environmental parameters.

## Author Contributions

**Conceptualization:** Paweł Chmura, Hongyou Liu, Marcin Andrzejewski, Jan Chmura, Marek Konefał.

**Data curation:** Edward Kowalczuk, Marek Konefał.

**Formal analysis:** Paweł Chmura, Hongyou Liu, Marek Konefał.

**Funding acquisition:** Jan Chmura, Andrzej Rokita.

**Investigation:** Paweł Chmura, Hongyou Liu.

**Methodology:** Paweł Chmura, Hongyou Liu, Marcin Andrzejewski, Jan Chmura, Marek Konefał.

**Project administration:** Paweł Chmura.

**Resources:** Paweł Chmura, Jan Chmura, Edward Kowalczuk.

**Software:** Edward Kowalczuk, Marek Konefał.

**Supervision:** Marcin Andrzejewski, Andrzej Rokita.

**Validation:** Paweł Chmura, Andrzej Rokita.

**Visualization:** Hongyou Liu, Marek Konefał.

**Writing – original draft:** Paweł Chmura, Hongyou Liu.

**Writing – review & editing:** Paweł Chmura, Marcin Andrzejewski, Jan Chmura, Marek Konefał.

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
