## [Decision Letter · Decision Letter 0]

7 Jan 2021

PONE-D-20-32150

Is there meaningful influence from situational and environmental factors on the physical and technical activity of elite football players? Evidence from the data of 5 consecutive seasons of the German Bundesliga

PLOS ONE

Dear Dr. Konefał,

Thank you for submitting your manuscript to PLOS ONE. After careful consideration, we feel that it has merit but does not fully meet PLOS ONE’s publication criteria as it currently stands. Therefore, we invite you to submit a revised version of the manuscript that addresses the points raised during the review process.

We look forward to receiving your revised manuscript.

Kind regards,

Yih-Kuen Jan, PhD

Academic Editor

PLOS ONE

Journal Requirements:

2. Please include your tables as part of your main manuscript and remove the individual files. Please note that supplementary tables (should remain/ be uploaded) as separate "supporting information" files

3. Thank you for including your ethics statement:  "This study maintains the anonymity of the players following data protection law, is conducted in compliance with the Declaration of Helsinki and was approved by the local Board of Ethics in Wrocław. No 20/2017. Written.".   

"no"

We note that one or more of the authors are employed by a commercial company: Football Club, Hannover 96, Robert-Enke-Str. 1, Hannover 30169, Germany

(2) Please also provide an updated Competing Interests Statement declaring this commercial affiliation along with any other relevant declarations relating to employment, consultancy, patents, products in development, or marketed products, etc.  

Please respond by return email with an updated Funding Statement and Competing Interests Statement and we will change the online submission form on your behalf.

6. We note that you have indicated that data from this study are available upon request. PLOS only allows data to be available upon request if there are legal or ethical restrictions on sharing data publicly. For information on unacceptable data access restrictions, please see http://journals.plos.org/plosone/s/data-availability#loc-unacceptable-data-access-restrictions.

Reviewers' comments:

Reviewer's Responses to Questions

**Comments to the Author**

1. Is the manuscript technically sound, and do the data support the conclusions?

Reviewer #1: Yes

Reviewer #2: Yes

2. Has the statistical analysis been performed appropriately and rigorously? 

Reviewer #1: Yes

Reviewer #2: Yes

3. Have the authors made all data underlying the findings in their manuscript fully available?

Reviewer #1: Yes

Reviewer #2: Yes

4. Is the manuscript presented in an intelligible fashion and written in standard English?

Reviewer #1: Yes

Reviewer #2: Yes

5. Review Comments to the Author

Reviewer #1: 1. Please correct the abbreviation WGBT (Wet Bulb Globe Temperature)

2. Please provide the ethical reference number from the local Board of Ethics (pg5)

3. Please justify that the sample size in this study was enough based on the holistic approach to monitoring and larger sample size as stated on the page 3

4. Please state how the number of 779 players with 21,971 individual match observations were considered in the analysis (CD n=6187, FB n=4806, CM=4798, WM n=3545, FW n=2635). Were there any changes in the players' positions in each match or other matches?

5. Please consider to apply the right justify for the values in the Table 2

6. It is more meaningful to have the minimum and maximum values in the Figures. eg the highest and lowest temperature, humidity etc.

Reviewer #2: line 109: WGBT (Wet Bulb Globe Temperature) index should be written as " Wet Bulb Globe Temperature ( WGBT) index

Lines 142-145 should be moved to the descriptive part of the results section. These abbreviations should have been written in full before being used (CD , FB , CM 143 ,WM , FW)

Figures 2 to 4 should be made more viewable for reading by keeping at most 3 graphs per row.

The authors should avoid section numberings

6. PLOS authors have the option to publish the peer review history of their article (what does this mean?). If published, this will include your full peer review and any attached files.

Reviewer #1: No

Reviewer #2: No

---

## [Author Response · Author response to Decision Letter 0]

19 Jan 2021

Reviewer 1

In response to your comments in the review of the manuscript PONE-D-20-32150 "Is there meaningful influence from situational and environmental factors on the physical and technical activity of elite football players? Evidence from the data of 5 consecutive seasons of the German Bundesliga", we would like to inform you about corrections made and augmentations provided. 

Reviewer #1: 1. Please correct the abbreviation WGBT (Wet Bulb Globe Temperature)

The correction has been made.

2. Please provide the ethical reference number from the local Board of Ethics (pg5)

The correction has been made.

3. Please justify that the sample size in this study was enough based on the holistic approach to monitoring and larger sample size as stated on the page 3

a) We thank the Reviewer for this remark. In the opinion of the authors, a research sample containing over 21,000 authorizes a holistic approach. The analysis divided into players positions shows that each one contains over 2,600 observations, which is definitely more than in other publications of similar type. For example, on the evolution of football

- Barnes C, Archer DT, Hogg B, Bush M, Bradley PS. The evolution of physical and technical performance parameters in the English Premier League. Int J Sports Med. 2014 Dec; 35(13): 1095-100.

4. Please state how the number of 779 players with 21,971 individual match observations were considered in the analysis (CD n=6187, FB n=4806, CM=4798, WM n=3545, FW n=2635). Were there any changes in the players' positions in each match or other matches?

We thank the Reviewer for this remark. Yes, the players’ positions were sometimes changed from match to match. But according to the data provider, within one match, the playing position of each player is assigned by his club at the start of the game, and would not have been altered (Definitionskatalog Offizielle Spieldaten – Bundesliga website - https://s.bundesliga.com/assets/doc/10000/2189_original.pdf) 

5. Please consider to apply the right justify for the values in the Table 2

The correction has been made.

6. It is more meaningful to have the minimum and maximum values in the Figures. eg the highest and lowest temperature, humidity etc.

We thank the Reviewer for this remark. The data presented in the Figures were effect data. The minimum and maximum values of temperature and humidity were descriptive data, which have been added into the revised table 2.

We would like to express our sincere thanks to the Reviewers for their time and valuable comments on our study. We trust that the corrections and revisions made will improve the editorial value and content of our study and allow us have our work published in your renowned journal. 

Sincerely yours,

The Authors

Reviewer 2

In response to your comments in the review of the manuscript PONE-D-20-32150 "Is there meaningful influence from situational and environmental factors on the physical and technical activity of elite football players? Evidence from the data of 5 consecutive seasons of the German Bundesliga", we would like to inform you about corrections made and augmentations provided. 

Reviewer #2: line 109: WGBT (Wet Bulb Globe Temperature) index should be written as " Wet Bulb Globe Temperature (WBGT) index

The correction has been made.

Lines 142-145 should be moved to the descriptive part of the results section. These abbreviations should have been written in full before being used (CD , FB , CM 143 ,WM , FW)

The correction has been made. 

Figures 2 to 4 should be made more viewable for reading by keeping at most 3 graphs per row.

The authors tried to generate figures according to the reviewer's suggestions, but then the figures were less readable - it is better if all 5 items are on the horizontal axis and the next parameters are arranged in lines one below the other than vice versa. Due to the analysis of five player positions, many factors and the desire not to increase the number of figures, the authors after an in-depth discussion, decided to leave the figures in their current form. 

The authors should avoid section numberings

The correction has been made.

We would like to express our sincere thanks to the Reviewers for their time and valuable comments on our study. We trust that the corrections and revisions made will improve the editorial value and content of our study and allow us have our work published in your renowned journal. 

Sincerely yours,

The Authors

---

## [Decision Letter · Decision Letter 1]

15 Feb 2021

Is there meaningful influence from situational and environmental factors on the physical and technical activity of elite football players? Evidence from the data of 5 consecutive seasons of the German Bundesliga

PONE-D-20-32150R1

Dear Dr. Konefał,

We’re pleased to inform you that your manuscript has been judged scientifically suitable for publication and will be formally accepted for publication once it meets all outstanding technical requirements.

Kind regards,

Yih-Kuen Jan, PhD

Academic Editor

PLOS ONE

Additional Editor Comments (optional):

Reviewers' comments:

Reviewer's Responses to Questions

**Comments to the Author**

1. If the authors have adequately addressed your comments raised in a previous round of review and you feel that this manuscript is now acceptable for publication, you may indicate that here to bypass the “Comments to the Author” section, enter your conflict of interest statement in the “Confidential to Editor” section, and submit your "Accept" recommendation.

Reviewer #1: All comments have been addressed

Reviewer #2: All comments have been addressed

2. Is the manuscript technically sound, and do the data support the conclusions?

Reviewer #1: Yes

Reviewer #2: Yes

3. Has the statistical analysis been performed appropriately and rigorously? 

Reviewer #1: Yes

Reviewer #2: Yes

4. Have the authors made all data underlying the findings in their manuscript fully available?

Reviewer #1: Yes

Reviewer #2: Yes

5. Is the manuscript presented in an intelligible fashion and written in standard English?

Reviewer #1: Yes

Reviewer #2: Yes

6. Review Comments to the Author

Reviewer #1: previous comment was "Please consider to apply the right justify for the values in the Table 2". However it was done as "left justify". The reason for the left justify is to have the same alignment of decimal points in the table.

For example:

10.80±0.85

31.40±1.50

24.40±7.40

Thank you

Reviewer #2: The authors have addressed all my comments sufficiently and i have no other comments to be addressed.

7. PLOS authors have the option to publish the peer review history of their article (what does this mean?). If published, this will include your full peer review and any attached files.

Reviewer #1: No

Reviewer #2: No

---

## [Editor Report · Acceptance letter]

23 Feb 2021

PONE-D-20-32150R1 

Is there meaningful influence from situational and environmental factors on the physical and technical activity of elite football players? Evidence from the data of 5 consecutive seasons of the German Bundesliga 

Dear Dr. Konefał:

I'm pleased to inform you that your manuscript has been deemed suitable for publication in PLOS ONE. Congratulations! Your manuscript is now with our production department. 

Kind regards, 

on behalf of

Dr. Yih-Kuen Jan 

Academic Editor

PLOS ONE